# Discovery of the Genus *Anapleus* Horn, 1873 from Cretaceous Kachin Amber (Coleoptera: Histeridae) [note 1]

**DOI:** 10.3390/insects13080746

**Published:** 2022-08-19

**Authors:** Rixin Jiang, Michael S. Caterino, Xiangsheng Chen

**Affiliations:** 1Institute of Entomology, Guizhou University, Guiyang 550025, China; 2The Provincial Special Key Laboratory for Development and Utilization of Insect Resources of Guizhou, Guizhou University, Guiyang 550025, China; 3The Provincial Key Laboratory for Agricultural Pest Management of Mountainous Region, Guizhou University, Guiyang 550025, China; 4Department of Plant & Environmental Sciences, Clemson University, Clemson, SC 29634, USA

**Keywords:** histeroidea, anapleini, new species, fossil species

## Abstract

**Simple Summary:**

Historically, researchers have suggested different resolutions of the basal relationships of the family Histeridae based on various datasets and methods of phylogenetic analysis. Phylogenetic analyses combining extant and fossil forms will doubtlessly shed further light on its early evolution. The present study describes the first fossil *Anapleus* species from mid-Cretaceous Kachin amber. This new discovery enriches the fossil record of histerid beetles and has important implications for efforts to understand their early evolutionary history.

**Abstract:**

For the first time, an extant histerid genus *Anapleus* Horn, 1873 is described from a specimen found in mid-Cretaceous Kachin amber. *Anapleus kachinensis* sp. nov. Although the genus *Anapleus* has not been precisely defined by synapomorphies, the new species shares numerous features with extant species while differing in comparatively few external characteristics. *Anapleus kachinensis* represents the first record of an extant histerid genus from Cretaceous deposits and provides further evidence of the ancient origin of the genus.

## 1. Introduction

The small histerid beetle genus *Anapleus* Horn, 1873 currently comprises 16 extant species and is widely distributed from Europe and Asia to North and Central America [1]. Members of *Anapleus* share a number of characteristics: (1) body small, oval and convex; (2) frontal stria of head absent, surface obliquely convex in front of eyes and feebly depressed on longitudinal median line; (3) labrum transverse, with a pair (or more) of setiferous punctures; (4) antenna stout, scape oblong and stout, pedicel somewhat elongate and thick, club consisting of the three apical antennomeres, of which the sutures are distinct; (5) pronotum transverse, sides usually strongly convergent to apices; (6) elytra usually coarsely punctate, sometimes strigose apically, dorsal striae absent, lateral margins carinate, apex of elytra truncate; (7) propygidium transverse and nearly vertical, pygidium curved downwards, prosternal keel quadrate and broad, truncate to shallowly emarginate at base, suture between lobe and keel indistinct; (8) antennal fissure longitudinal, deep and situated along prosternal lobe and process; (9) carinal striae of prosternal keel deeply impressed; (10) mesosternum short and transverse, meso-metasternal suture distinct and crenate, metasternum coarsely and densely punctate [2]. Though none of these characteristics has been hypothesized as an explicit autapomorphy of this genus, there has been little question of its monophyly in the literature to date. Yet, apart from isolated descriptions, and faunal treatments by Ohara [2] for Japan and Olexa [3] for the broader Palearctic, the genus has received very little systematic attention.

Previous phylogenetic research on the family as a whole has suggested a basal position of the genus *Anapleus* in the family Histeridae [4,5], based largely on apomorphies of most other Histeridae that *Anapleus* and a few other genera seem to lack. However, a sparse fossil record for this family leaves much to learn about their early evolution. This situation has improved over the past few years, especially through a series of works on Kachin amber inclusions [6,7,8,9,10]. However, most of these fossils have been highly distinct from extant taxa, and difficult to place even in a subfamily.

In this paper, we report the first *Anapleus* species from Upper Cretaceous Kachin amber. Based on a three-dimensional and well-preserved specimen, *Anapleus kachinensis* sp. nov. is described and figured. The new finding provides further evidence of the ancient origin of the genus *Anapleus* and will provide a valuable calibration point for the assessment of early histerid evolution.

## 2. Materials and Methods

The fossil beetle described here is embedded in amber from the Hukawng Valley, northern Myanmar (26°21′33.41″ N, 96°43′11.88″ E) [11,12,13]. The age of Kachin amber is widely accepted as the earliest Cenomanian (98.79 ± 0.62 Ma) [14].

The amber piece was cut using a handheld engraving tool and polished with sandpapers of different grain sizes and rare earth polishing powder to help observe and photograph the beetle.

Habitus images were taken using a Canon 5D Mark IV digital camera with an MP-E 65 mm f/2.8 1–5X macro lens. A Godox MF12 flash was used as the light source. Images of the morphological details were taken using a Canon 5D Mark IV camera with a Mitutoyo Plan NIR 10 lens or a Nikon SMZ25 stereoscopic microscope with a Nikon DS-Ri2 camera.

Montages were produced in Zerene Stacker Version 1.04. All images were modified and grouped in Adobe Photoshop CS5 Extended.

Morphological terms in this paper follow Ohara (1994) [2].

The amber specimen in this study is deposited in the Institute of Entomology, Guizhou University, Guiyang, China (GUGC).

## 3. Systematic Palaeontology

Order Coleoptera Linnaeus, 1758.

Superfamily Histeroidea Gyllenhal, 1808.

Family Histeridae Gyllenhal, 1808.

Subfamily Histerinae Gyllenhal, 1808.

Tribe Anapleini Olexa, 1982.

Genus *Anapleus* Horn, 1873.

***Anapleus kachinensis*** Jiang, Caterino & Chen sp. nov.

(Figure 1, Figure 2 and Figure 3).

urn:lsid:zoobank.org:pub:131A5A42-CE3C-455C-97D8-75B46CE4F54B.

**Etymology.** The specific epithet refers to the type locality, Burma; adjective.

**Material.** Holotype, labeled ‘GUGC-FOSSIL-0002, *Anapleus kachinensis* Jiang, Caterino & Chen sp. nov., Lowermost Cenomanian, mid-Cretaceous (ca. 99 Ma), from near the village of Tanai, Hukawng Valley, northern Myanmar.’ (GUGC), the amber piece is flat and rounded; its size is 10 mm in length, 7.5 mm in width, and 1 mm in height. The head of the specimen is covered with a large bubble, which partly obscures the dorsal view.

**Diagnosis.** Body, elongate oval, surface covered with different sized, large round punctures; vestiges of elytral striae visible at the elytral base; protibia with distinct apical spurs; postcoxal lines of the metaventrite and first abdominal ventrite more strongly curved than extant species of the genus. Head transverse; pronotum widest at base, basal margin strongly curved. Scutellum quite small, near triangular, apex rounded. Both inner side and outer side of posterior tibiae with rows of long spines, rows on inner side shorter than on outer side, about half the width of tibiae.

**Description.** Body (Figure 1A,B) small, oval, black with antennae, mandibles, and reddish brown legs, surface finely covered with different sized, large round punctures, body length 1.29 mm.

Head transverse, head length 0.22 mm, width 0.32 mm, eyes small, weakly convex. Antenna (Figure 2B) with 11 antennomeres; club formed by three apical antennomeres; scape long, broad, pedicel cylindrical, about three times as long as antennomere III; antennomere III–V shorter, longer than wide; antennomere VI–VIII intestiniform; antennomere IX–XI forming a club, strongly transverse, surface covered with dense short setae, apex of antennomere XI rounded. Mandible (Figure 2A) strong, finely curved, inner side fimbriate along basal margin, with small tooth near apex. Labrum (Figure 2D) wide, lateral margins curved, covered with several long setae, apical margin straight, without setae; dominant discal setae not evident. Maxillae lost; prementum of labium present, wide, narrowed to apex, other parts of labium lost. Gular sutures (Figure 2D) fused.

Pronotum transverse, pronotum length 0.42 mm, width 0.69 mm, widest at base, and narrowed from base to apex, dorsal surface finely covered with large round punctures. Anterior margin slightly emarginate, lateral margins cambered and weakly margined, basal margin strongly curved, subangulate.

Prosternal keel (Figure 2E) emarginate at base, with widely separated, parallel carinal striae (Figure 2E) extending onto base of prosternal lobe, keel flat between; prosternal lobe (Figure 2E) distinctly arcuate, surface finely covered with large round punctures. Antennal fissures (Figure 2C) longitudinal. Prosternal keel striae (Figure 2E) almost continuous with marginal stria of the prosternal lobe. Procoxae widely separated.

Scutellum (Figure 1A) quite small but could be observed, near triangular, with round apex, surface without punctures. Elytra (Figure 1A) wider than long, elytra length 0.65 mm, width 0.77 mm, widest near basal 1/3, disc distinctly convex, with lateral margin carinate, surface finely covered with different sized and rounded punctures, vestiges of elytral striae visible as scratch-like marks near base.

Mesoventrite projecting anterad, weakly angulate, wide, with distinct lateral striae; metaventrite (Figure 3A) surface smooth, covered with different sized and rounded punctures, punctures much smaller at disc and middle area near basal margin, and much larger at other areas of metaventrite; lateral metaventral disk with postmesocoxal stria strongly recurved to mesepimeron. Median sulcus longitudinal, thin and shallow, extending from base to apex of metaventrite.

Abdomen (Figure 3A) with five visible sternites, surface finely covered with large punctures. Sternite III longest, slightly shorter than combined length of sternites IV–VII, posterior margin concave, with postmetacoxal stria recurved at side nearly to anterolateral corner; sternite IV shorter than combined length of sternites V–VII; sternite V slightly longer than sternite VI; sternite VI about as long as sternite VII.

Profemur broad, flat, punctate on inner posterior margin; protibiae (Figure 3C) narrow, slightly expanded at middle, outer margin with row of long spines, inner side with row of thin spines extending from apex to near middle, major and minor apical spurs present; tarsal groove along outer margin of protibia, protarsus cannot be observed. Meso- and metafemora similar, rather narrow, slightly expanded at middle. Meso-(Figure 3D) and metatibiae (Figure 3B) weakly expanded at apical 1/2, with few stronger spines on outer margin, larger number of short, thin spines on inner margin; apical spurs small, but distinct. Tarsi 5-5-5, basal four tarsomeres with only a pair of apicoventral setae; pretarsal claws paired, thin, separate.

## 4. Discussion

The new species exhibits some unusual characteristics compared with extant members of the genus *Anapleus*. Its longer and more oval body form contrasts with the typically broadly rounded form of extant *Anapleus* species. More significantly, it possesses a few likely plesiomorphic characteristics [4] that appear to be lost in extant *Anapleus*, such as the distinct (especially pro-) tibial spurs and the relatively prominent spines on the outer margins of the tibiae. The postcoxal striae (both meso- and meta-) are more strongly curved in *A. kachinensis* than in any extant species that we have been able to study. However, based on other combined characteristics, here we still place the new species in the genus *Anapleus*.

*Anapleus kachinensis* exhibits some characteristics that may prove informative with respect to relationships among extant *Anapleus*. Extant species vary significantly in the degree of emargination in the base of the prosternal keel, with most Asian and Palearctic species (e.g., *A. cyclonotus* Lewis, 1892) having the keel only very weakly emarginate [2,3]. In the American *A. marginatus* (LeConte), 1853, the keel is distinctly emarginate, as it is in *A. kachinensis*. The Asian species also generally have long series of fine setae on the venters of the basal four tarsomeres, unlike *A. marginatus* and *A. kachinensis*, which have only an apical pair on each tarsomere [2].

There are, nonetheless, several important characteristics that are obscured in the single specimens of *A. kachinensis* that would help assess its phylogenetic position. The antennal club of *Anapleus* is unique among Histeridae in having distinctly incised annuli, appearing to completely separate the antennomeres of the club. Since the club antennomeres in all other Histeridae appear fused with only superficial annuli, it would be very useful to be able to assess this characteristic confidently in *A. kachinensis*. However, one is missing, and the other is at a difficult angle and partly obscured by a large bubble. The form of the protibia is also at a difficult angle to properly observe. Extant *Anapleus* have a unique marginal protarsal groove, bounded by fine setae on inner and outer edges. That of *A. kachinensis* is similar, but also appears to be less well defined. The obscurement of the front of the head also hinders assessment of the shape of the antennal scape, the openings of the antennal insertions, and the setation of the labrum, all potentially useful in placement and polarization of characteristics across Anapleini [2].

Historically, researchers have suggested different resolutions of the basal relationships of the family Histeridae. Crowson [15] and Kryzhanovskij [16] suggested that the subfamily Abraeinae, Niponiinae, Trypanaeinae, and Trypeticinae might represent relatively early diverging lineages. The first explicit phylogenetic analysis of the group by Slipiński and Mazur [17] resolved the genus *Niponius* Lewis, 1885 as sister to the remainder of the family, implying the cylindrical body form, and a life of hunting under tree bark as the basal habit of the family. However, later phylogenetic work combining morphological characters of larvae and adults and 18S rDNA sequence data suggested that, instead, histerid beetles with ovate body form might be basal, as represented by the genera *Anapleus*, *Onthophilus* Leach, 1817, or *Dendrophilus* Leach, 1817 [4]. This hypothesis is so far consistent with a series of newly reported fossils [7,8,10,18,19,20]. The new finding of the first extinct record of the genus *Anapleus* in Cretaceous amber is also consistent with this idea, although the discovery of additional fossils and more thorough phylogenetic analyses combining extant and fossil forms will doubtlessly continue to shed further light on early histerid evolution.

## Figures and Tables

**Figure 1 insects-13-00746-f001:**
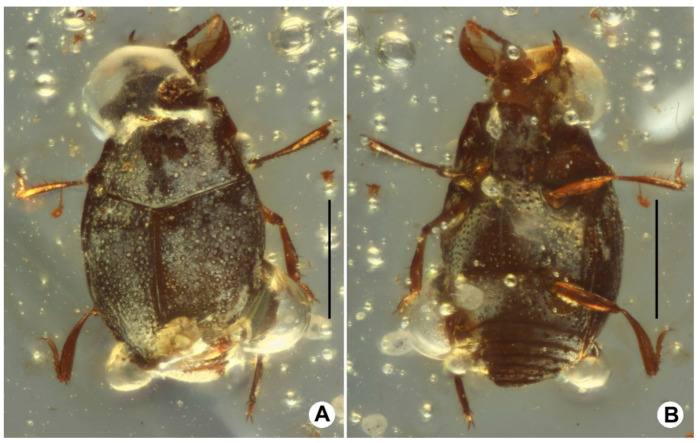
Habitus of *Anapleus kachinensis* sp. nov., holotype, GUGC-FOSSIL-0002. (**A**) Dorsal view. (**B**) Ventral view. Scalebars: 0.5 mm.

**Figure 2 insects-13-00746-f002:**
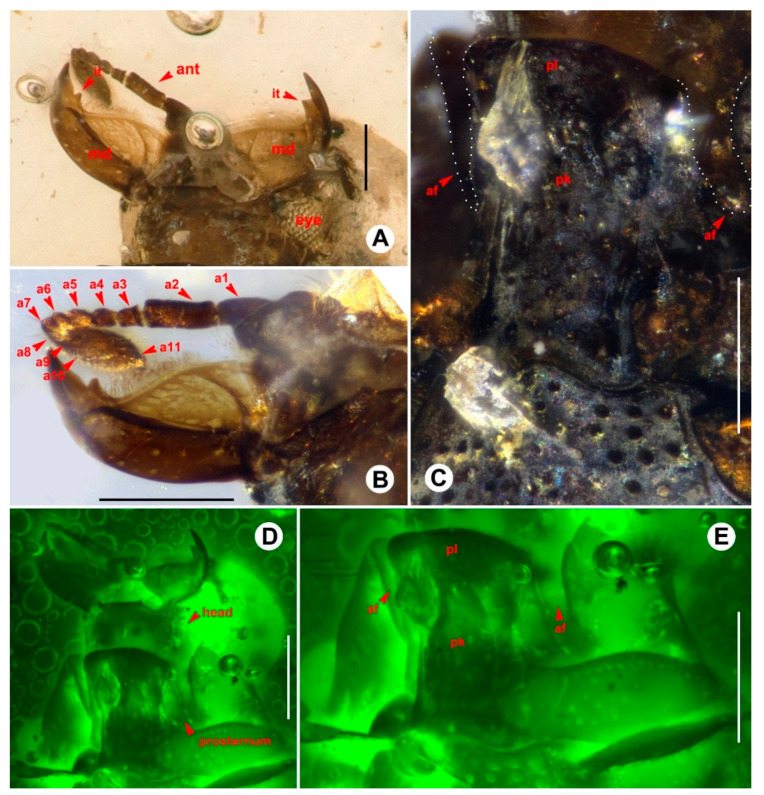
Diagnostic features of *Anapleus kachinensis* sp. nov., holotype, GUGC-FOSSIL-0002. (**A**) Mandible. (**B**) Antenna. (**C)** Prosternum. (**D**) Head and prosternum, under green fluorescence. (**E**) Prosternum, under green fluorescence. Scale bars: 0.1 mm. Abbreviations: ant, antenna; a1–11, antennomere 1–11; md, mandible; it, inner tooth; af, antennal fissures; pk, prosternal keel; pl, prosternal lobe.

**Figure 3 insects-13-00746-f003:**
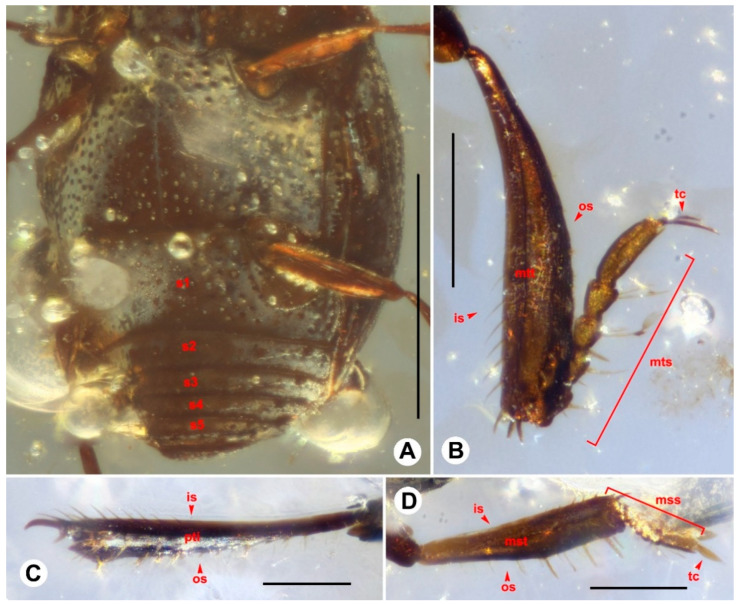
Diagnostic features of *Anapleus kachinensis* sp. nov., holotype, GUGC-FOSSIL-0002. (**A**) Abdomen. (**B**) Metatibia. (**C**) Protibia. (**D**) Mesotibia. Scale bars: 0.5 mm in (**A**), 0.1 mm in (**B**–**D**). Abbreviations: s1–5, sterna III–VII; is, inner side; os, outer side; pti, protibia; mst, mesotibia; mss, mesotarsus; mtt, metatibia; mts, metatarsus; tc, tarsal claw.

## Data Availability

No new data were created or analyzed in this study. Data sharing is not applicable to this article.

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
