# Peer review of "Discovery of the Genus Anapleus Horn, 1873 from Cretaceous Kachin Amber (Coleoptera: Histeridae)†"

_insects, 2022, doi:10.3390/insects13080746_

Round 1

Reviewer 1 Report

The study described the oldest Anapleus species from the mid-Cretaceous Kachin amber, which is important for the extinct fauna of Coleoptera. The description and illustrations are proper; however, the following issues should be addressed before further consideration.

1. Simple Summary: line 2, rewrite the sentence into ‘...by different ways of phylogenetic analysis...’.

2. Simple Summary: line 4, ‘Burmese amber’ of the whole manuscript needs to be replaced as ‘Kachin amber’ since there are more than one amber deposits in Burma.

3. Abstract: line 3, ‘been precisely defined by synapomorphies’, I think the generic definition should derive from autapomorphies instead of synapomorphies.

Line 4, ‘differs in relatively few’? The sentence is not complete, differs in relatively few what?

‘This is the first extant genus of Histeridae’ is misleading, better to rewrite as ‘first extinct record of Anapleus...’.

4. Introduction: line 3, Anapleus should be italic. Check similar mistakes in the next text.

Line 16, the use of ‘synapomorphy’ is incorrect, please check this word throughout the manuscript, e.g., synapomorphy of A and B; autapomorphy of A.

5. Materials and Methods: please add the references of terminology if there are any.

6. Systematic Palaeontology: to avoid homonym, please consider changing another specific name, burmensis has been used in too many insects including Coleoptera.

7. In all figures, it would be much better to mark the diagnostic structures such as scutellum in the figures considering the wide readership of the journal.

8. Discussion: line 1, ‘exhibits some unusual characters among extant members’, among? The new species is not extant. Rewrite the sentence.

Line 2, ‘Its more elongate body form’, I’m afraid the body form of the fossil is rounded rather than elongate, please choose other more distinct characters for the comparison.

Line 3, ‘it possesses a few likely plesiomorphic characters’. Is there any literature mentioning these characters as plesiomorphic? Or are these plesiomorphic characters newly proposed by the authors? Please clarify the reason why they are plesiomorphic characters.

Second paragraph, line 4-5, provide author and year for all scientific names when they appear for the first time. Add information for A. cyclonotus Lewis, A. marginatus (LeConte), and other names in the manuscript.

Second and third paragraphs, add references for all the comparative discussion with extant species.

Last paragraph, add both author and year information for Niponius and other genera. Check other genera and species names in the manuscript.

Line 11, ‘The finding of Anapleus as the first extant histerid genus’, rewrite this sentence according to previous comment 3.

Line 1-11 are reviews of previous studies; this part should be placed into the introduction section.

9. Last but not least, there are numerous linguistic mistakes and misuses which are not easy to understand especially for general readers. Please find someone to polish the language, which will largely improve the readability of this paper.

Author Response

Dear Reviewer:

Much thanks for all your advice of manuscript.

Best wishes.

Jiang

Reviewer 2 Report

I do not have major comments, only some small corrections. I included them directly in the text file attached here.

Author Response

(The authors gave the same response as above.)

Reviewer 3 Report

the manuscript is well prepared and there are only few things needed corrections. In general I recommend to the authors to erect a new genus with dagnostic defined by the co-authors in their discussion. I think the co-author can add also shorter tarsi, and particularly tarsomere 1.

Author Response

(The authors gave the same response as above.)
